# A Comparative Elemental Analysis of Espresso Coffee from Poland and Portugal

**DOI:** 10.3390/foods14030426

**Published:** 2025-01-28

**Authors:** Pawel Konieczyński, Kinga Seroczyńska, Marek Wesolowski, Edgar Pinto, Cristina Couto, Ana Cunha, Rui Azevedo, Agostinho Almeida

**Affiliations:** 1Department of Analytical Chemistry, Medical University of Gdansk, Gen. J. Hallera 107, 80-416 Gdansk, Poland; pawelkon@gumed.edu.pl (P.K.); marek.wesolowski@gumed.edu.pl (M.W.); 2REQUIMTE/LAQV, ESS, Polytechnic of Porto, Rua Dr. António Bernardino de Almeida 400, 4200-072 Porto, Portugal; ecp@ess.ipp.pt; 3LAQV/REQUIMTE, Department of Chemical Sciences, Faculty of Pharmacy, University of Porto, Rua de Jorge Viterbo Ferreira 228, 4050-313 Porto, Portugalruiazevedo43@gmail.com (R.A.); 4Associate Laboratory i4HB-Institute for Health and Bioeconomy, University Institute of Health Sciences-CESPU, 4585-116 Gandra, Portugal; 5UCIBIO-Applied Molecular Biosciences Unit, Forensics and Biomedical Sciences Research Laboratory, University Institute of Health Sciences (1H-TOXRUN, IUCS-CESPU), 4585-116 Gandra, Portugal

**Keywords:** espresso coffee, macrominerals, trace elements, nutritional value, food safety

## Abstract

A comparative elemental analysis of espresso coffee from Poland and Portugal was carried out. Using an ICP-MS analytical procedure, samples collected from public cafes in Poland and Portugal (n = 60 and n = 44, respectively) were studied for their macromineral and trace element content. To evaluate the contribution of water to the final composition of the beverage, paired samples (i.e., collected from the same locations) of drinking water were also analysed. The mineral profile of the coffee espresso samples was quite similar: Mg > P > Ca > Rb > Mn > B > Zn > Cu > Sr > Ba > Ni > Pb > Cs > Mo > Sn > Cd > Sb > Tl for samples from Poland and Mg > P > Ca > Rb > B > Mn > Zn > Sr > Cu > Ni > Ba > Cs > Pb > Mo > Sn > Sb > Cd > Tl for samples from Portugal. For most of the elements, the espresso samples showed much higher levels than the water used in its preparation. The two most notable exceptions were Ca and Sr, where the elements present in the coffee came mainly from the water. The contribution of coffee espressos to the daily intake of essential elements seems to be reduced. Other non-essential elements like Ni (median = 81.0 µg/L and 86.8 µg/L for Polish and Portuguese espresso, respectively) and Pb (median = 14.3 µg/L and 4.43 µg/L, respectively) were observed in significant amounts in the coffee espresso samples analysed in this study. These elements have been shown to leach from coffee machines in other studies. More studies are necessary to confirm these results.

## 1. Introduction

Coffee is one of the most popular beverages in the world, brewed from roasted and finely powdered beans (berry seeds) of plants from the *Rubiaceae* family, genus *Coffea* [1]. There is extensive evidence to suggest an association between moderate coffee consumption and a range of health outcomes, from improved cognitive function and athletic performance to reduced risk of various diseases [2]. In two umbrella reviews of the evidence from meta-analyses, an association was found between coffee consumption and a reduction in the relative risk of all-cause mortality, cardiovascular mortality, and cardiovascular disease. Coffee consumption was also associated with a decreased risk of several specific cancers (breast, colorectal, colon, endometrial, and prostate cancer), Parkinson’s disease, and type 2 diabetes [3,4].

There are many different processes for preparing coffee beverages, but “espresso” has become one of the most popular in Europe [5]. In 2020, *per capita* coffee consumption in Portugal was 5.05 kg, while in Poland it was only 2.23 kg [6]. In comparison, in Luxembourg coffee consumption reached 25.3 kg/person [6].

Although procedures may vary somewhat, the Italian Espresso National Institute describes the espresso-making procedure as passing water at high temperature (86–92 °C) and high pressure (8–10 bar) through coffee powder (6.5–7.5 g of roasted and ground coffee beans) for 20–30 s to produce about 25 mL of beverage [7]. The extraction of chemical substances from coffee powder is very efficient, so their concentration in coffee (beverage) is also very high [8]. Variables related to the coffee powder (e.g., plant species of bean origin, and roasting and grinding processes) affect the appearance, smell, and taste of the beverage. Additionally, specific variables related to powder extraction, such as water temperature and composition, extraction time, and pressure and powder/water ratio, decisively affect the quality of the final drink [8].

Regarding its composition, coffee is a complex product, containing more than 1000 substances with different biological activity, including macronutrients and micronutrients such as vitamins, phenolic compounds, and various minerals [9]. During the beverage preparation process, water-soluble components are extracted, including chlorogenic acids, caffeine, nicotinic acid, soluble melanoidins, and volatile hydrophilic compounds [10]. Part of the lipid fraction, although not water-soluble, also reaches the beverage due to high temperature and pressure. This polyphasic nature of the beverage and the concentration of volatile and non-volatile compounds determine its physicochemical and sensory properties [10].

Although the mineral content of coffee is only about 5% (*w*/*w*), it also affects its organoleptic properties. Knowledge of the content of macrominerals and essential trace elements and the possible presence of potentially toxic trace elements is important in terms of evaluating the nutritional value and safety of the beverage [11]. Potentially toxic elements include metals that at trace levels are essential, because they are involved in the normal functioning of living organisms (e.g., Cu, Mn, Co, and Zn), but at high levels can become toxic. On the other hand, elements such as Pb, Cd, As, and Hg are not necessary for the normal activity of living organisms and are toxic, even in trace concentrations [9,12].

Due to the importance of essential and toxic elements for human health and the widespread coffee consumption in many countries, it becomes important to monitor their concentration in coffee. This type of study is of high importance for public health entities, providing important data for the assessment of the oral exposure level to multiple elements (essential and non-essential). These types of data are also important to the coffee industry interested in improving the safety/quality of products. The present study aimed to contribute to answering the basic question “What do we drink when we drink a cup of coffee?” regarding mineral elements. A comparative study was carried out on espresso coffee samples collected in two different European locations: Gdansk region (Poland; 60 samples) and Porto metropolitan area (Portugal; 44 samples). To evaluate the real contribution of coffee powder to the mineral element content of the beverage, paired analysis of the water used in beverage preparation was also carried out.

## 2. Materials and Methods

### 2.1. Coffee and Water Samples Collection

In this study, a total of 104 paired coffee and tap water samples were analysed, 60 collected in Northern Poland (Gdańsk, Gdynia, and Elbląg area) and 44 in Northern Portugal (Porto metropolitan area). Espresso samples (5–10 mL) were taken (with a plastic pipette, into polyethylene tubes), directly from the beverage served in cafes, restaurants, and gas stations. To evaluate the real contribution of coffee (powder) to the final composition of the beverage, water samples (15 mL, into polyethylene tubes) were also collected in the same locations.

All samples were then taken to the laboratory. The samples collected in Poland were kept frozen until shipping to Portugal (under the same conditions), where they were thawed before analysis. The Portuguese samples were kept refrigerated until analysis.

### 2.2. Reagents and Materials

Calibration standards were prepared by appropriate dilution of the 10 mg/L multielement standard solution PlasmaCAL SCP-33-MS (SCP Science, Baie-d’Urfé, QC, Canada) and 1000 mg/L Mg, P, and Ca single-element standard solutions, all from Fluka (L’Isle d’Abeau Chesnes, France). A 100 µg/L internal standards (IS) solution was prepared by diluting four single-element standard solutions (1000 mg/L): Indium ICP Standard Solution, Praseodymium ICP Standard Solution, Terbium ICP Standard Solution, and Lutetium ICP Standard Solution, all also from Fluka.

All solutions were prepared with ultrapure water obtained from an Arium^®^ pro water purification system (Sartorius, Göettingen, Germany). High-purity concentrated HNO_3_ (≥69% *w*/*w*, TraceSELECT^®^, Fluka) was used to acidify (at 2% *v*/*v*) both the samples and standard solutions. To prevent contamination, all labware (tubes and volumetric flasks) was decontaminated by immersion in a 10% HNO_3_ solution for at least 24 h, followed by abundant rinsing with ultrapure water and drying in dust-protected conditions.

### 2.3. Samples Preparation

Sample preparation for analysis was the same for water and coffee samples. After the samples were brought to room temperature and homogenised, 1 mL of each sample was taken and 2 mL of 10% (*v*/*v*) HNO_3_ was added. The prepared mixture was left for 24 h at room temperature for digestion. Then, 7 mL of ultrapure water was added, followed by centrifugation at 4500 rpm for 1 min (Heraeus Megafuge 16 Centrifuge, Thermo Scientific, Hemel Hempstead, UK). From this 1:10 dilution, 2 mL was taken, which was mixed with 2 mL of 20 µg/L IS solution (in 2% HNO_3_). The same procedure was used for the preparation of the sample blanks.

### 2.4. Determination of Elemental Concentrations

Elemental analysis was performed by Inductively Coupled Plasma Mass Spectrometry (ICP-MS) using an iCAP™ Q instrument (Thermo Fisher Scientific, Bremen, Germany), equipped with a baffled cyclonic spray chamber (Peltier-cooled), a MicroMist™ nebuliser (Glass Expansion, Port Melbourne, Australia), a standard quartz torch, and nickel sample, and skimmer cones. High-purity argon (99.9997%; Gasin II, Matosinhos, Portugal) was used as nebuliser and plasma gas. The ICP-MS instrument was operated under the following conditions: radiofrequency power—1550 W; cool gas flow—14.0 L/min; auxiliary gas flow—0.80 L/min; nebuliser gas flow—1.20 mL/min; and dwell time—10 ms.

The following *m/z* ratios were monitored for analytical determination: ^11^B, ^26^Mg, ^31^P, ^43^Ca, ^55^Mn, ^60^Ni, ^65^Cu, ^66^Zn, ^75^As, ^82^Se, ^85^Rb, ^88^Sr, ^98^Mo, ^111^Cd, ^118^Sn, ^121^Sb, ^133^Cs, ^137^Ba, ^205^Tl, and ^208^Pb. The following *m/z* ratios were monitored as IS: ^89^Y, ^141^Pr, ^159^Tb, and ^175^Lu. Limits of detection (LDs) were calculated as the concentration corresponding to 3.3xSD of the sample blanks. For analytical quality control, a high-matrix drinking water reference material (EnviroMAT, SCP Science, Quebec, Canada) was used. Analyte recovery ranged from 99% to 112%.

### 2.5. Data Analysis

Statistical analysis of the data was performed using SPSS Statistics 25.0 software (IBM Corp., Armonk, NY, USA). For mathematical calculations, results below the LD were imputed as LD/square root of 2, a procedure commonly used in similar cases [13]. Statistical significance was set at *p* < 0.05. Variables distribution normality was assessed using the Kolmogorov–Smirnov test. All elements presented a distribution significantly different to the normal distribution (*p* < 0.05). To evaluate the statistical significance of the differences in elemental concentrations between Portuguese and Polish samples, the non-parametric Mann–Whitney test was used.

## 3. Results

A panel of 20 elements was analysed in the 104 paired espresso and water samples (60 from Poland and 44 from Portugal). The LD ranged from 0.026 µg/L for Tl to 128 µg/L for Ca, as shown Table 1, which also indicates the percentage of samples that presented concentrations below the LD for the different samples analysed.

Figure 1 and Appendix A summarise the comparison of the selected elements’ concentration in coffee samples from Poland and Portugal. The mineral profile of the samples collected in Poland presented the following order of abundance: Mg > P > Ca > Rb > Mn > B > Zn > Cu > Sr > Ba > Ni > Pb > Cs > Mo > Sn > Cd > Sb > Tl. For the Portuguese samples, the profile was quite similar: Mg > P > Ca > Rb > B > Mn > Zn > Sr > Cu > Ni > Ba > Cs > Pb > Mo > Sn > Sb > Cd > Tl. Samples from Poland showed higher concentrations of most elements (*p* < 0.05). The exceptions were Rb (*p* = 0.389), Ni (*p* = 0.747), Cs (*p* = 0.129), Mo (*p* = 0.252), Sn (*p* = 0.054), Sb (*p* = 0.322), and Tl (*p* = 0.623). The concentration of As and Se was lower than the LD in, respectively, 67% and 100% of the coffee samples (Table 1), and they were therefore excluded from the comparative statistical analysis.

In addition to coffee powder, water is the other essential ingredient for making espresso coffee, and the most important from a quantitative point of view [14]. Figure 2 and Appendix A show the concentration of selected elements in water samples from Poland and Portugal. The mineral profile of samples collected in Poland was Ca > Mg > Sr > Zn > P > B > Cu > Ba > Ni > Mn > Pb > Rb > Mo > Sn > Sb. For the Portuguese samples, the profile found was Ca > Mg > Sr > P > Zn > Cu > Ba > B > Ni > Mn > Rb > Pb > Mo > Sb > Sn.

Notably, Polish water samples were much more mineralised. Except for Sb, which was higher in the Portuguese samples (*p* < 0.001), and for Rb, where the difference did not reach statistical significance (*p* = 0.708), all the remaining elements were found in significantly higher concentrations in the samples from Poland (*p* < 0.05). The concentrations of As, Se, Cd, Cs, and Tl were lower than the LD in 69%, 100%, 65%, 66%, and 83% of the samples (Table 1), respectively, and therefore they were not considered in the comparative statistical analysis.

Figure 3 shows the percentage contribution of water to the final concentration of the analysed elements in the espresso samples. As can be seen, water contributes significantly to the final concentration of Ca and some trace elements. The Ca concentration in Polish water represented 92% of the concentration found in espresso samples, while in the case of Portugal it corresponded to only about 53%. Similarly, the Sr content in water from Poland and Portugal corresponded to 110% and 85% of the content in coffee samples from Poland and Portugal, respectively. The content of Zn, Mo, Cd, Sn, Ba, and Pb in water corresponded to roughly 10% to 60% of the content found in coffee in both countries. For B, Mg, P, Mn, Ni, Rb, and Cs, the contribution of water was much less significant, generally less than 10%.

## 4. Discussion

In the present study, the mineral composition of coffee espresso samples collected in Poland and Portugal was studied. To evaluate the real contribution of coffee powder to the content of the beverage, the composition of the water used to prepare it was also analysed. The content of most elements in coffee samples was significantly different between Poland and Portugal, but this was mainly associated with the mineral composition of the drinking water used to prepare the beverage, which was also significantly different.

The study focused on a broad panel of elements relevant to assessing the nutritional value and safety of the beverage. Some minerals are needed in large dietary doses to ensure normal body function (e.g., Ca and Mg), while others, such as Cu, Zn, and Mo, are only needed in small daily amounts. Dietary Reference Values (DRV) is a generic term for a set of nutrient reference values that indicate the dose of a nutrient that should be ingested regularly to maintain health in an otherwise healthy individual (or population) [15]. In the EU, these values are defined and occasionally updated by the EFSA, and are used by policymakers to issue recommendations on nutrient intake to consumers. In the case of non-essential/toxic elements, dietary intake must be kept below specified limits, due to the risk of adverse effects related to excessive exposure. The U.S. Agency for Toxic Substances and Disease Registry (ATSDR) has set a minimal risk level (MRL) for oral exposure for several different elements. The contribution of consuming two cups of coffee espresso to the DRV of essential elements and MRL of non-essential elements is shown in Table 2.

Magnesium acts as a cofactor for more than 300 enzymatic reactions, regulating several biochemical mechanisms in the human body [28]. The median (P25–P75) Mg concentration in the Polish samples was 243 (186–344) mg/L, while the median for the Portuguese samples was 171 (131–195) mg/L. The highest Mg content reported by Oliveira et al. [29] was 152.4 ± 7.0 mg/100 g in Brazilian espresso samples, while the average Mg content in commercial pure origin espresso coffees from 13 different regions (39 espresso samples prepared with coffee from Africa, South and Central America, Asia, and Oceania) was 118.5 ± 24.9 mg/100 g, which is ~10 times higher than the median Mg concentration in coffee espresso observed in the present study (assuming a coffee espresso density of ~1 g/mL). Another study comparing five different brewing methods of an espresso blend—100% arabica (AeroPress, drip, espresso machine, French press, and simple infusion) in Poland found a concentration of ~85 mg/L for the espresso samples [30]. However, espresso samples were prepared with 17 g of coffee powder and 250 mL of water [30], which may explain the lower Mg concentrations observed. According to the review by Olechno et al. (2021), the Mg content of several different coffee brews varied between 2.15 and 14.9 mg/100 mL [9]. The median (P25–P75) Mg concentration in the analysed Polish and Portuguese water samples was 10.7 (9.19–12.7) mg/L and 5.04 (4.47–5.86) mg/L, respectively, representing only 3–5% of the Mg concentration in espresso samples. The extraction of Mg from powdered coffee seems to be moderately efficient (48–55%) [31]. The contribution of the Mg content of the water samples analysed in this study to the espresso Mg concentration is much reduced. The differences in the Mg concentration in Polish and Portuguese espresso samples might be explained by differences in coffee composition, particle size of the coffee powder, or extraction conditions (e.g., water temperature, pressure, espresso volume, etc.) [8]. The EFSA’s Adequate Intake (AI) levels for Mg are 350 mg/day and 300 mg/day for adult men and women, respectively [15]. Therefore, espresso coffee, according to the results obtained in the present study, does not constitute a significant source of this mineral. Even for samples from Poland, two cups of espresso coffee (2 × 25 mL) represent less than 4% of the AI (Table 2).

Phosphorus is an essential nutrient for both skeletal and non-skeletal tissues, and is particularly crucial for energy production [32]. The median (P25–P75) P concentration was 211 (147–344) mg/L in the Polish samples and 167 (142–200) mg/L in the Portuguese samples. Oliveira et al. [29] reported an average total P content of 480.2 ± 130.0 mg/100 g in espresso, with the highest content in Kenyan coffee (673.2 ± 43.4 mg/100 g) and the lowest in Cuban coffee (270.3 ± 1.9 mg/100 g), which are a lot higher than the coffee espresso Ca concentration observed in the present study. Janda et al. (2020) observed a P concentration of 5.6 mg/100 mL for the coffee espresso samples, but the espresso samples were prepared with 17 g of coffee powder and 250 mL of water [30]. The median (P25–P75) P concentration in Polish and Portuguese water samples was 127 (110–297) µg/L and 88.4 (26.1–172) µg/L, respectively, which corresponds to <0.1% of P concentration in espresso samples. Similar to Mg, the extraction of P from ground coffee is high (45–46%) [31]. The differences in P concentration in espresso samples from Poland and Portugal might be due to differences in coffee composition, particle size of the coffee powder, or extraction conditions (e.g., water temperature, pressure, espresso volume, etc.) [8]. Considering the AI of 550 mg/day for adults [15], two cups of coffee espresso (25 mL) do not represent a significant source of P (<2% of the AI; Table 2).

Calcium plays important roles in virtually every cell in the human body [33]. The Ca Population Reference Intake (PRI) set by the EFSA for adults (≥25 years) is 1000 mg/day [15]. Calcium was the mineral with the third highest median concentration in espresso samples from both Poland and Portugal. The median (P25–P75) concentration in the Polish samples was 80.8 (63.1–109) mg/L, and in the Portuguese samples 52.2 (38.5–60.9) mg/L. The mean Ca content in espresso samples reported by Oliveira et al. [29] was 27.8 ± 11.0 mg/100 g. In another study, a Ca concentration of 25.7 mg/L was found in espresso samples, which is much lower than in our study, but the beverage was prepared with 17 g of coffee powder and 250 mL of water [30]. In the review by Olechno et al. (2021), the Ca content in different coffee brews ranged from 1.38 to 3.49 mg/100 mL [9]. The median (P25–P75) Ca concentration in Polish and Portuguese water samples was 76.7 (64.9–90.1) mg/L and 26.7 (20.2–30.4) mg/L, respectively, around 90% and 50% of the median Ca content in Polish and Portuguese espresso, respectively. The efficiency of Ca extraction during the preparation of coffee beverages seems to be much more variable (10–65%) [31] and is lower when the water used has a high concentration of Ca, possibly due to the formation of complexes with polyphenols and other organic constituents, and precipitation as CaCO_3_ [9]. In the present study, the Ca extraction from coffee was more efficient in the samples from Portugal, which used water samples with a lower Ca concentration. The intake of two cups of espresso provides <1% of the Ca PRI (Table 2). Therefore, espresso is not a significant dietary source of Ca, not differing significantly from simple water intake in the case of Poland (Table 2).

Manganese is an essential component of a few human metalloenzymes [15]. In the present study, the median (P25–P75) Mn concentration in the Polish and Portuguese espresso samples was 1713 (1173–2762) µg/L and 815 (658–1118) µg/L, respectively. On the other hand, the median (P25–P75) Mn concentration in water samples was 3.31 (1.97–6.72) µg/L and 1.91 (0.715–3.74) µg/L, respectively. This means that the high Mn content found in espresso coffee comes almost entirely from Mn present in coffee powder, which is extracted during beverage preparation. The extraction of Mn from ground coffee seems to be lower than other elements (19–39%) due to the formation of covalent bonds between Mn and other coffee constituents [31]. Water has a negligible contribution. The Mn AI for adults set by the EFSA is 3000 µg/day [15]. Thus, the consumption of two cups of coffee espresso could provide around 1.5% and 3% of the AI (according to our results for the Portuguese and Polish samples, respectively; Table 2). Oliveira et al. [29] found a mean Mn content of 0.450 ± 0.173 mg/100 g in espresso samples, with the highest content recorded in coffee from India (0.703 ± 0.004 mg/100 g), and the lowest in samples from China (0.056 ± 0.001 mg/100 g). A similar study on Polish coffee beverages found a Mn concentration in espresso of around 0.5 mg/L [30], much lower than that obtained in our study, but, as already mentioned, espresso samples were prepared with only 17 g of coffee powder to 250 mL of water.

Zinc plays a crucial role in several physiological processes, such as cell growth and development, metabolism, and cognitive, reproductive, and immune system function [34]. The median (P25–P75) Zn concentration in the espresso samples analysed was 660 (336–1202) µg/L (samples Poland) and 276 (156–557) µg/L (samples from Portugal). Considering the EFSA PRI for Zn (7.5–12.7 mg/day and 9.4–16.3 mg/day for adult men and women, respectively, depending on the daily intake of phytate) [15], it can be concluded that espresso consumption does not contribute significantly to the daily intake of this trace element (Table 2). Jarošová et al. [35] determined the Zn content in coffee bean and found values ranging from 1.71 ± 0.09 mg/kg (Kenyan samples) to 7.12 ± 0.64 mg/kg (Indian samples). The aforementioned Polish study found a Zn concentration of around 0.23 mg/L for espresso samples (prepared with 17 g of coffee powder and 250 mL of water) [30]. In the review by Pohl et al. (2013), the concentration of Zn in powdered and instant coffee brew ranged from 4.1 to 29.0 mg/kg [31], while, in a more recent review by Olechno et al. (2021), Zn concentration ranged from 17 µg/100 mL to 23.5 mg/100 mL in coffee brewed in an espresso machine [36]. Assuming a density of ~1 g/mL, the Zn concentrations observed in the present study seem to be much lower than what was observed in most of the other studies mentioned. The concentration of Zn varied greatly in both coffee espresso and water samples. The median (P25–P75) concentration of Zn in water samples from Poland and Portugal was 187 (69.5–470) and 63.1 (29.9–128) µg/L, respectively. The water seems to contribute ~25% of the Zn observed in the coffee espresso samples (Figure 3 and Appendix A), but the high variability seems to indicate that coffee powder composition and differences in coffee espresso preparation procedure are responsible for the high variability observed. Additionally, the extraction of Zn from ground coffee seems to be highly variable (8.6–62%), depending on the formation of complexes with polyphenols, alkaloids, or other compounds [31,36].

Copper is an essential micronutrient required for electron transfer processes [15]. The median (P25–P75) Cu concentration in Polish espresso samples was 551 (112–1220) µg/L, while in Portuguese samples was only 178 (72.1–492) µg/L. In water samples, the median (P25–P75) Cu concentration was 64.2 (27.1–127) µg/L and 20.5 (12.0–41.2) µg/L, respectively, representing 10–14% of the Cu content in espresso. Copper does not seem to leach in significant amounts from coffee machines [37], and the efficiency of Cu extraction during the preparation of coffee beverages is very reduced (2.6–8.2%) [31], so the wide difference between the samples from Poland and Portugal might be due to differences in coffee powder composition, particle size, or the coffee extraction procedure. Similar to Zn, Cu seems to form complexes with other components of coffee [36], which hinders its extraction during the preparation of coffee espresso. The AI set by the EFSA for Cu is 1.6 mg/day and 1.3 mg/day for adult men and women, respectively [15]. Therefore, espresso does not represent a significant source of dietary Cu (Table 2). Jarošová et al. [35] reported the highest Cu content in coffee bean samples from Kenya (19.25 ± 0.77 mg/kg), and the lowest in samples from Honduras (16.61 ± 0.52 mg/kg). The Cu concentration in the Polish study [30] (espresso prepared with 17 g of coffee powder and 250 mL of water) was around 80 µg/L. In the review by Pohl et al. (2013), the Cu content in powdered and instant coffee brews ranged from 0.7 to 3.2 mg/kg [31], while a more recent review by Olechno et al. (2021) described Cu levels ranging from 2.36 to 8.5 µg/100 mL in coffee brewed in espresso machines [36].

In humans, Mo is mainly found in the active site of enzymes that are involved in redox reactions, where the pterin-bound Mo ion undergoes a cycling process between the Mo(VI) and Mo(IV) oxidation states [38]. The median (P25–P75) Mo concentration in espresso samples was 2.56 (1.28–4.27) µg/L and 1.97 (0.889–4.13) µg/L, for Polish and Portuguese samples, respectively. To our knowledge, no other study has analysed Mo concentration in espresso samples. The median (P25–P75) Mo concentration in water samples was 1.18 (0.832–1.71) µg/L (Poland) and 0.417 (0.233–0.545) µg/L (Portugal), which corresponds, respectively, to 60% and 14% of the observed concentration of Mo in the espresso samples analysed in this study. Curiously, by subtracting the water Mo concentration from the Mo concentration in the espresso samples, the coffee seems to provide ~1.4 µg/L of the Mo concentration. The EFSA has set a Mo AI for adults of 65 µg/day [15]. The contribution of espresso consumption to this intake is not relevant (Table 2). A study in Turkey that analysed 12 different samples of green coffee beans found a mean Mo content of 0.12 µg/g [11].

Strontium is a non-essential alkaline earth metal chemically similar to Ca [19]. Pharmacological doses of Sr-ranelate are used for osteoporosis treatment, although only in very specific cases, due to the increased cardiovascular risk associated with its use [39,40]. Food and drinking water are the main sources of exposure to Sr [24]. The median (P25–P75) Sr content in Polish espresso samples was 273 (206–376) µg/L, while in Portuguese samples it was 189 (134–265) µg/L. For water samples, the median (P25–P75) Sr concentration was 274 (224–567) µg/L (Poland) and 178 (138–183) µg/L (Portugal), which corresponds to approximately all the Sr present in the espresso samples. Considering the chemical similarities of Ca and Sr, it is possible that Sr also forms complexes with polyphenols and other organic constituents [9], which reduces its extraction during the preparation of coffee beverages. In fact, the median (P25–P75) concentration of Sr observed in Polish water samples corresponded to 110% (71–229%) of the Sr concentration in Polish coffee espresso samples (Appendix A). This seems to indicate that the ground coffee used to prepare the coffee espresso was able to retain a very significant portion of the water Sr. The ATSDR has set a MRL for oral exposure of 2.0 mg/kg/day (140 mg/day for a 70 kg individual) [19]. Thus, according to the results obtained in this study, espresso does not appear to be a significantly different source of Sr than mere consumption of drinking water (Table 2). Janda et al. [30], for the study of espresso samples prepared with 17 g of coffee powder and 250 mL of water, reported an Sr concentration of 734 µg/L.

Rubidium is a non-essential alkali metal [41]. The median (P25–P75) Rb concentration in Polish and Portuguese espresso samples was 4984 (3913–6671) µg/L and 4665 (3460–6453) µg/L, respectively. To our knowledge, no other study has analysed Rb content in espresso samples. The concentration of Rb in water samples was comparatively negligible—the median (P25–P75) concentration was 1.62 (1.15–2.43) µg/L (Poland) and 1.70 (1.44–1.83) µg/L (Portugal), which corresponds to <0.1% of the Rb content found in the espresso samples. It can therefore be concluded that coffee powder is very rich in Rb, and the element is easily extracted during the preparation of espresso. The authors could not find any evidence regarding the extraction efficiency of Rb, but K and Na (also alkali metals) have an extraction efficiency of 71–88% and 36–48% from ground coffee, respectively, during the preparation of coffee beverages.

Boron is not considered essential for humans, although some evidence suggests that B is bioactive and beneficial for several physiological processes [42]. The B content of powdered and instant coffee brews seems to range from 3.3 to 11.8 mg/kg [31], much higher than the B concentration in the coffee espresso samples analysed in the present study. The median (P25–P75) concentration found in the Polish and Portuguese samples analysed was 1545 (1084–2133) µg/L and 1234 (971–1503) µg/L, respectively. In water samples, the median (P25–P75) concentration of B was 105 (59.2–211) µg/L and 6.97 (4.93–11.0) µg/L, respectively. The contribution of water to the final content of B in the espresso was therefore very low, around 7% in the Polish samples and <1% in the Portuguese samples. In fact, after subtracting the B concentration in each water sample from the paired coffee espresso sample, the difference between Polish and Portuguese samples is very small—1291 (910–1783) and 1226 (971–1481), respectively. As for Rb, coffee powder is very rich in B, and the element is easily extracted during espresso preparation. The ATSDR has established an oral MRL of 0.2 mg/kg/day for B (14 mg/day for a 70 kg adult) [16]. Therefore, consuming two cups of coffee espresso would expose a 70 kg adult to <1% of the MRL (Table 2).

Nickel is an abundant transition metal, essential for plants but not humans [17]. On the contrary, the role of Ni as a causative agent of a wide variety of diseases has been the subject of intense research in recent years [43]. The median (P25–P75) Ni concentration in the espresso samples analysed was quite similar: 81.0 (45.3–133) µg/L in samples from Poland, and 86.8 (58.0–121) µg/L in samples from Portugal. Várady et al. [44] found a Ni content of 0.63 ± 0.07 mg/kg and 0.48 ± 0.03 mg/kg in green coffee beans from Colombia and Nicaragua, respectively. Similarly, a Turkish study that analysed a total of 12 different samples of green coffee beans found a mean Ni content of 0.54 µg/g [11]. A review by Pohl et al. (2013) observed that the Ni content in powdered and instant coffee brews ranged from <0.002 to 1.35 mg/kg. Nickel is widely used as a component of metal alloys and stainless steel [17], and its leaching in considerable amounts has been evidenced in coffee machines, especially in portafilter espresso machines and after descaling [37]. Variations in the ground coffee composition and coffee espresso preparation procedure, in addition to the ubiquitous presence in the materials used to store, process, and transport coffee, might explain the variability in coffee espresso Ni concentration observed in the present study and the available literature. The median (P25–P75) water Ni concentration in water samples was also quite similar: 5.58 (4.08–9.62) µg/L in Polish samples and 4.61 (1.09–6.55) µg/L in Portuguese samples, which corresponds to 5–7% of the Ni concentration found in the analysed espresso samples. The contribution of water to the Ni concentration observed in the coffee espressos samples is reduced, but the methodology used in the present study does not allow the identification of the source of Ni (coffee powder vs. coffee machine). The ATSDR has not established an oral MRL for Ni [17], but the EFSA has derived a Tolerable Daily Intake (TDI) of 13 µg/kg/day, which corresponds to 91 µg/day for a 70 kg individual [18]. According to the data obtained in the present study, consuming two cups of coffee espresso would represent an exposure to ~5% of the TDI.

Barium is an alkaline earth metal widely used as an X-ray contrast agent in medical imaging [24]. The median (P25–P75) Ba concentration in espresso samples was 146 (96.6–229) µg/L and 79.8 (51.2–108) µg/L, for the Polish and Portuguese samples, respectively. The ATSDR has established an oral MRL for Ba of 0.2 mg/kg/day (14 mg/day for a 70 kg adult). The intake of two cups of coffee espresso would provide <0.1% of the MRL (Table 2). To our knowledge, there are no reports on the Ba content in espresso. The median (P25–P75) Ba concentration in Polish and Portuguese water samples was 23.7 (19.1–36.8) µg/L and 17.6 (14.5–20.4) µg/L, respectively, which represents 17–22% of the Ba concentration found in espresso samples analysed in the present study. The extraction of Ca, another alkali earth metal, is influenced by the water Ca concentration and the content of Ca in ground coffee [9]. It is possible that the Ba extraction from ground coffee is similarly affected.

Lead is a toxic trace element associated with a wide range of negative effects on the human body [26]. No MRL has been set by the ATSDR for Pb because even the lowest blood Pb levels observed in epidemiological studies (≤5 µg/dL) are associated with significant adverse effects (e.g., decreased cognitive function in children) [26]. However, the EFSA established a Benchmark Dose Level associated with a 10% increase in the risk of nephrotoxicity (BMDL_10_) of 0.63 µg/kg/day [27], which corresponds to 44.1 µg/day for a 70 kg individual. In the present study, the median (P25–P75) Pb concentration in espresso samples was 14.3 (2.49–34.0) µg/L in samples from Poland and 4.43 (1.99–15.6) µg/L in samples from Portugal. The intake of two cups of coffee espresso, according to the data obtained in the present study, would provide <2% of the BMDL_10_ set by the EFSA (Table 2). Winiarska-Mieczan et al. [45] found an average Pb content of 49.6 µg/kg in dry coffee powder. These coffee powder samples were used to prepare coffee infusions (6 g of coffee powder was poured into 100 mL of drinking water at 95–100 °C and the suspension was filtered through a Whatman filter after 10 min). On average, 94% of Pb passed from the coffee powder to the infusion [45]. Müller et al. [37] showed that Pb can also be leached from coffee machines, sometimes at levels potentially concerning for human health, particularly in portafilter espresso machines and after descaling. The median (P25–P75) Pb concentration in water samples from Poland and Portugal was 2.15 (1.58–3.89) µg/L and 0.574 (0.254–1.37) µg/L, respectively, which means that only about 21% and 10% of the Pb content found in espresso samples comes from the water used in its preparation. The differences in Pb concentration in espresso samples from Poland and Portugal might be due to differences in coffee composition, particle size of the coffee powder, extraction conditions (e.g., water temperature, pressure, espresso volume, etc.) [8], or even due to differences in coffee machine build and maintenance.

Caesium is a non-essential, low-toxic alkaline metal with some industrial applications [23]. To our knowledge, there are no data on the Cs content in espresso samples. The median (P25–P75) Cs concentration was 8.44 (5.84–16.5) µg/L for Polish espresso samples, and 12.4 (8.00–17.2) µg/L for Portuguese samples. In water, the concentration of Cs was <LD (0.055 µg/L) in 66% of the samples. The Cs content in water therefore showed a negligible contribution (<1.0%) to the Cs concentration found in the espresso samples analysed. The differences in Cs concentration in espresso samples from Poland and Portugal might be due to differences in coffee composition, particle size of the coffee powder, and extraction conditions (e.g., water temperature, pressure, espresso volume, etc.) [8]. The ATSDR has not established an oral MRL for Cs [23].

Tin is a non-essential element and has been used to make metal containers (cans) for storing food and beverages since the 19th century [46]. The median (P25–P75) Sn concentration in espresso samples was 0.442 (0.173–1.65) µg/L for Polish samples and 1.27 (0.320–4.22) µg/L for Portuguese samples. To our knowledge, no other study has analysed Sn concentration in espresso samples. A mean content of 9.84 µg/g was found in the analysis of 12 different green coffee samples [11]. The median (P25–P75) Sn concentration in the water samples was 0.155 (0.104–0.750) µg/L (samples from Poland) and 0.110 (<0.056–0.247) µg/L (samples from Portugal). This represents, respectively, about 35% and 20% of the Sn concentration found in the espresso samples. The differences in Sn concentration in espresso samples from Poland and Portugal might be due to differences in coffee composition, particle size of the coffee powder, extraction conditions (e.g., water temperature, pressure, espresso volume, etc.) [8], or even due to differences in coffee machine build and maintenance. A thin layer of Sn is sometimes added to the metallic components of food-processing equipment, milk cans, and kitchen implements [47]. The Sn observed in the coffee samples, albeit in very small concentrations, could have leached from the equipment used during coffee bean storage, grinding, and espresso preparation. The ATSDR has set a MRL for oral exposure of 0.3 mg/kg/day (21 mg/day for a 70 kg individual) [21]. The intake of two cups of coffee espresso does not seem to contribute significantly to the daily oral exposure to Sn (Table 2).

Arsenic and Se were <LD (2.78 µg/L and 8.55 µg/L, respectively) in more than two thirds of the espresso samples. The concentration of Cd, Sb, and Tl was <0.2 µg/L in most espresso samples, and lower than the LD (0.068 µg/L, 0.044 µg/L and 0.026 µg/L, respectively) in 43%, 36%, and 46% of samples, respectively. Given that these elements are highly toxic to humans [20,22,25], the results of this study support the safety of espresso beverage in both geographical regions studied. The ATSDR has established an oral MRL for Cd and Sb of 0.1 µg/kg/day and 0.6 µg/kg/day, respectively, but not for Tl [20,22,25]. The consumption of two cups of coffee espresso exposes the individual to 0.1% or less of the oral MRL of Cd and Sb (Table 2).

The present study provides an interesting picture of the concentration of several different elements in coffee espresso available to the general population in two different regions of Europe. The number of collected samples was reduced and does not necessarily present an accurate picture of the elemental composition of coffee espresso in Poland and Portugal, but provides, nonetheless, interesting data for public health entities and industry partners regarding the quality and safety of coffee beverages. The additional collection and analysis of paired water samples provides additional detail regarding the source of the elements found in the final coffee beverage. Unfortunately, it was not possible to collect information related to the origin of the ground coffee or the espresso preparation procedure (e.g., water temperature and pressure, type of machine, final volume of the espresso, etc.), which would have added interesting data to the discussion of the results presented in this study.

## 5. Conclusions

A total of 104 pairs of espresso coffee and drinking water samples, collected in the same locations, 60 in Northern Poland and 44 in Northern Portugal, were analysed for the concentration of a panel of 20 elements (macro- and trace elements, essential and toxic). For most of the elements, the espresso samples showed much higher levels than those of the water used in its preparation, reflecting an extensive extraction from the coffee powder by the water at high pressure and temperature. The two most notable exceptions were Ca and Sr, where the element present in the coffee comes mainly from the water. The espresso samples from Poland were significantly more mineralised than those from Portugal, which was also observed for the drinking water. However, the difference was greater than could be explained by differences in the composition of the water, so that factors such as different compositions of the coffee powder or different espresso preparation processes (higher temperature, higher pressure, or smaller particle size of the powder) seem to be involved. In any case, given the low volume of beverage (espresso) usually consumed per day (1–3 cups, 25–75 mL each), its contribution to the daily intake of essential elements is low. On the other hand, potentially toxic elements were found at very low levels, within the limits for drinking water, demonstrating a good safety profile of the beverage in both countries.

## Figures and Tables

**Figure 1 foods-14-00426-f001:**
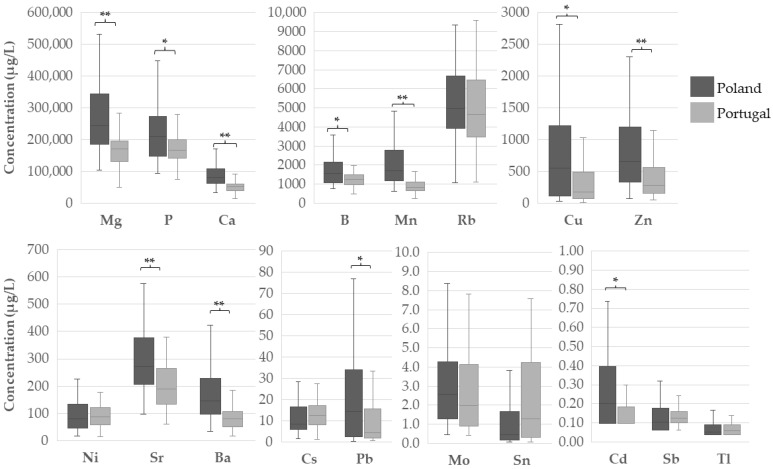
Concentration of macro- and trace elements in coffee samples from Poland and Portugal. (* *p* < 0.05; ** *p* ≤ 0.001; Mann–Whitney non-parametric test).

**Figure 2 foods-14-00426-f002:**
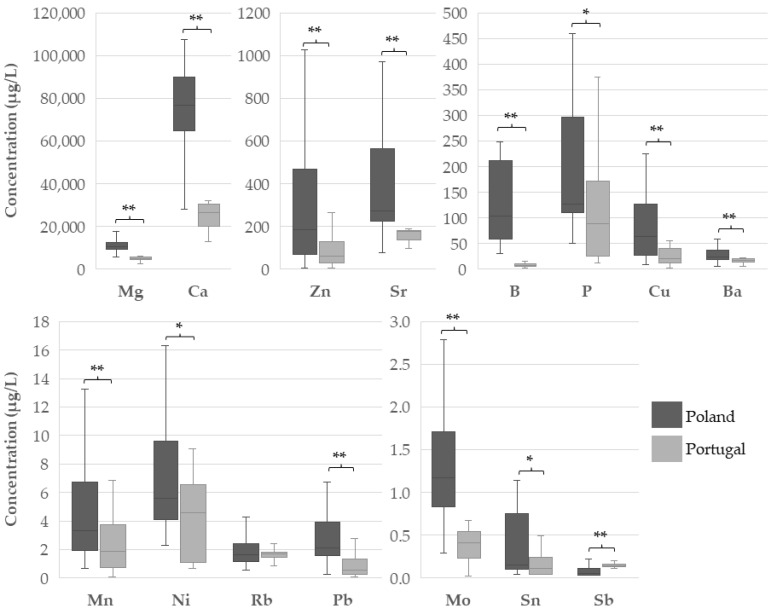
Concentration of macro- and trace elements in water samples from Poland and Portugal. (* *p* < 0.05; ** *p* ≤ 0.001; Mann–Whitney non-parametric test).

**Figure 3 foods-14-00426-f003:**
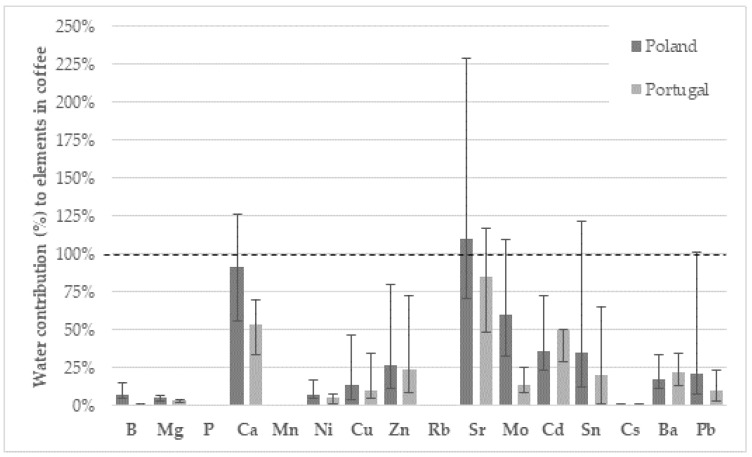
Median (P25–P75) contribution (%) of water to the concentration of macro- and trace elements in coffee samples from Poland and Portugal (dashed line represents the median concentration of each element in coffee espresso). The elements As, Se, Sb, and Tl were not included because a significant number of samples were <LD.

**Table 1 foods-14-00426-t001:** Limits of detection (LDs) and percentage (%) of samples below the LD.

Element	LD	Samples < LD		Element	LD	Samples < LD	
		Water	Espresso			Water	Espresso
Boron (B)	1.44	0%	0%	Rubidium (Rb)	0.072	0%	0%
Magnesium (Mg)	6.61	0%	0%	Strontium (Sr)	0.240	0%	0%
Phosphorus (P)	17.7	9%	0%	Molybdenum (Mo)	0.086	4%	0%
Calcium (Ca)	128	0%	0%	Cadmium (Cd)	0.068	65%	43%
Manganese (Mn)	0.141	3%	0%	Tin (Sn)	0.056	26%	8%
Nickel (Ni)	0.962	10%	0%	Antimony (Sb)	0.044	30%	36%
Copper (Cu)	0.634	0%	0%	Caesium (Cs)	0.055	66%	0%
Zinc (Zn)	1.33	0%	0%	Barium (Ba)	0.421	0%	0%
Arsenic (As)	2.78	69%	67%	Thallium (Tl)	0.026	83%	46%
Selenium (Se)	8.55	100%	100%	Lead (Pb)	0.100	4%	0%

**Table 2 foods-14-00426-t002:** Contribution (%) of two cups of coffee espresso to the DRV or MRL of each element.

Essential				Reference
Element	DRV (mg/day)	Poland	Portugal	
Mg	350	3.5%	2.4%	[15]
P	550	1.9%	1.5%	
Ca	1000	0.4%	0.3%	
Mn	3	2.9%	1.4%	
Cu	1.6	1.7%	0.6%	
Zn	7.5–12.7	0.3–0.4%	0.1–0.2%	
Mo	0.065	0.2%	0.2%	
Non-Essential				Reference
Element	MRL (mg/day)	Poland	Portugal	
B	14	0.6%	0.4%	[16]
Ni	0.91 *	4.5%	4.8%	[17,18]
Rb	na	-	-	-
Sr	140	<0.1%	<0.1%	[19]
Cd	0.007	0.1%	<0.1%	[20]
Sn	21	<0.1%	<0.1%	[21]
Sb	0.042	<0.1%	<0.1%	[22]
Cs	na	-	-	[23]
Ba	14	<0.1%	<0.1%	[24]
Tl	na	-	-	[25]
Pb	0.044 *	1.7%	0.5%	[26,27]

DRV—Dietary Reference Value; MRL—Oral Minimal Risk Level (for a 70 kg adult male; not available for all elements); na—not available. There are no MRL available for Ni, Rb, Cs, Tl, or Pb. * For Ni and Pb, a Tolerable Daily Intake (TDI) and a Benchmark Dose Level of 10% increased risk (BMDL_10_) of nephrotoxicity were used, respectively The elements As, Se, and Tl were not included because most of the samples were <LD.

## Data Availability

The original contributions presented in this study are included in the article/Appendix A. Further inquiries can be directed to the corresponding author.

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
