# Peer review of "A Comparative Elemental Analysis of Espresso Coffee from Poland and Portugal"

_foods, 2025, doi:10.3390/foods14030426_

Round 1
Reviewer 1 Report
Comments and Suggestions for Authors
The author compared the metal contents of coffee samples from two different regions (Poland and Portugal). A total of 104 samples were investigated, ensuring a broad scope of the study. Additionally, the influence of water on the metal content in coffee was also analyzed. The results indicated that the metal contents in the samples from the two regions were generally similar. Of course, some differences were also observed. Subsequently, the correlation between the concentrations of various metal ions and health was discussed.
The design of the paper is quite straightforward, resembling a purely data-driven research paper.
My only suggestion is whether the author should compare and discuss the detected metal ion levels with the specified metal ion content in coffee standards, such as national standards for coffee or standards set by relevant coffee companies.
The establishment of these standards is also based on the results of extensive investigations. It is possible that the author's investigation could provide some valuable references for updating these standards.
The main question addressed by the research: The author compared the metal contents of coffee samples from two different regions (Poland and Portugal). A total of 104 samples were investigated, ensuring a broad scope of the study. Additionally, the influence of water on the metal content in coffee was also analyzed. The results indicated that the metal contents in the samples from the two regions were generally similar. Subsequently, the correlation between the concentrations of various metal ions and health was discussed.
This topic is similar to those of many published literatures, such as these ones (Food Chemistry Volume 141, Issue 3, 1 December 2013, Pages 1956-1961, https://doi.org/10.1016/j.foodchem.2013.05.011 ). Compared with these published literatures, this paper focuses on coffee from this particular region, which holds certain scientific value especially for foods greatly influenced by regional differences (like coffee). Of course, from the perspective of applied value, this paper also contributes to the coffee industry, particularly the local coffee industry.
The design of the paper is quite straightforward, resembling a purely data-driven research paper. My suggestion is whether the author should compare and discuss the detected metal ion levels with the specified metal ion content in coffee standards, such as national standards for coffee or standards set by relevant coffee companies. A graph or table can be added here for further illustration. The establishment of these standards is also based on the results of extensive investigations. It is possible that the author's investigation could provide some valuable references for updating these standards.
References are appropriate.
Additional comments on the tables and figures: A graph or table can be added to compare and discuss the detected metal ion levels with the specified metal ion content in coffee standards.
Author Response
Response to Reviewer 1
The authors would like to thank the reviewer for the comments to the manuscript. Here are our replies to some of the questions and considerations pointed by the reviewer:
“The author compared the metal contents of coffee samples from two different regions (Poland and Portugal). A total of 104 samples were investigated, ensuring a broad scope of the study. Additionally, the influence of water on the metal content in coffee was also analyzed. The results indicated that the metal contents in the samples from the two regions were generally similar. Of course, some differences were also observed. Subsequently, the correlation between the concentrations of various metal ions and health was discussed.”
“This topic is similar to those of many published literatures, such as these ones (Food Chemistry Volume 141, Issue 3, 1 December 2013, Pages 1956-1961, https://doi.org/10.1016/j.foodchem.2013.05.011 ). Compared with these published literatures, this paper focuses on coffee from this particular region, which holds certain scientific value especially for foods greatly influenced by regional differences (like coffee). Of course, from the perspective of applied value, this paper also contributes to the coffee industry, particularly the local coffee industry.”
- The authors would like to thank the reviewer for the comments and mention that they elaborated further in the discussion section and compared the results of the study to other relevant literature with similar research topics.
“The design of the paper is quite straightforward, resembling a purely data-driven research paper. My suggestion is whether the author should compare and discuss the detected metal ion levels with the specified metal ion content in coffee standards, such as national standards for coffee or standards set by relevant coffee companies. A graph or table can be added here for further illustration. The establishment of these standards is also based on the results of extensive investigations. It is possible that the author's investigation could provide some valuable references for updating these standards.”
- The authors would like to thank the reviewer for the comments. The authors added a table (Table 2) with the contribution of drinking 2 cups of coffee espresso (with the data collected in this study) to the daily intake of the macro- and trace elements discussed in the manuscript. The authors also expanded on this subject in the text.
- The authors searched for literature or other relevant online sources of information regarding coffee standards or guiding values for the elemental content of coffee espresso. The authors also did not compare the results to legal limits because they were not able to find any for coffee espresso. However, the authors added a table (Table 2) with the contribution of drinking 2 cups of coffee espresso (with the data collected in this study) to the daily intake of the macro- and trace elements discussed in the manuscript. The authors also expanded on this subject in the text.
“References are appropriate.”
“Additional comments on the tables and figures: A graph or table can be added to compare and discuss the detected metal ion levels with the specified metal ion content in coffee standards.”
- A table (Table 2) was added with the contribution of drinking 2 cups of coffee espresso (with the data collected in this study) to the daily intake of the macro- and trace elements discussed in the manuscript.
Reviewer 2 Report
Comments and Suggestions for Authors
The article, "A Comparative Elemental Analysis of Espresso Coffee from Poland and Portugal," investigates the mineral content of espresso coffee and water samples using ICP-MS. This study offers valuable insights into the contributions of water and coffee powder to the mineral profile of espresso, emphasizing regional differences between Poland and Portugal. Here are my observations:
The introduction provides a thorough overview of coffee's popularity, brewing methods, and its nutritional and toxicological significance. While comprehensive, the discussion is overly general and lacks a focused research gap. Clearly articulate the research gap, such as a lack of comparative studies on elemental analysis of coffee from different regions. Streamline the introduction by removing redundant information about coffee brewing and focusing on the study’s objectives.
The methodology section provides detailed descriptions of sample collection, preparation, and analysis.
The results are presented in an organized manner, with data on the concentrations of 20 elements in coffee and water samples. However, the interpretation is overly descriptive and lacks a critical analysis of trends or implications. Discuss the variability within samples and provide error margins or confidence intervals for key measurements. Highlight the implications of potentially toxic elements, such as lead and nickel, and their sources (e.g., water, coffee powder, or machine leaching).
The discussion explores the mineral contributions of coffee powder and water, emphasizing the higher mineralization of Polish coffee. While informative, it lacks depth in exploring broader implications or limitations. Discuss how preparation variables (e.g., water temperature, pressure) might influence mineral extraction. Address study limitations, such as the small sample size or lack of control over preparation methods. Explore potential applications of the findings, such as optimizing brewing methods to minimize toxic element leaching. Compare findings with existing literature to contextualize the data.
The figures and tables effectively summarize elemental concentrations and contributions from water. Use error bars or confidence intervals to show variability in the data.
Author Response
Response to Reviewer 2
The authors would like to thank the reviewer for the comments to the manuscript. Here are our replies to some of the questions and considerations pointed by the reviewer:
“The article, "A Comparative Elemental Analysis of Espresso Coffee from Poland and Portugal," investigates the mineral content of espresso coffee and water samples using ICP-MS. This study offers valuable insights into the contributions of water and coffee powder to the mineral profile of espresso, emphasizing regional differences between Poland and Portugal. Here are my observations:”
“The introduction provides a thorough overview of coffee's popularity, brewing methods, and its nutritional and toxicological significance. While comprehensive, the discussion is overly general and lacks a focused research gap. Clearly articulate the research gap, such as a lack of comparative studies on elemental analysis of coffee from different regions. Streamline the introduction by removing redundant information about coffee brewing and focusing on the study’s objectives.”
- The authors tried to give the reader an overview of what coffee beverages are and to show that coffee espresso, in particular, is widely consumed on a regular basis. With the present study, we aimed to assess the concentration of several elements (essential and non-essential) in coffee espresso in order to provide valuable information for both public health entities and coffee industry. This type of data is useful, e.g., for studies assessing the overall intake of essential and non-essential elements or meta-analysis compiling/comparing the concentration of several constituents of coffee espresso. The additional analysis of paired water samples provides valuable information to better interpret the results. So, to make the objective of the study clearer, a sentence was added to the final paragraph of the introduction that now reads “Due to the importance of essential and toxic elements for human health and the widespread coffee consumption in many countries, it becomes important to monitor their concentration in coffee. This type of studies is of high importance for public health entities, providing important data for the assessment of the oral exposure level to multiple elements (essential and non-essential). This type of data is also important to the coffee industry interested in improving the safety/quality of their products. The present study aimed to contribute to answering the basic question "What do we drink when we drink a cup of coffee?" regarding mineral elements. A comparative study was carried out on espresso coffee samples collected in two different European locations: Gdansk region (Poland; 60 samples) and Porto metropolitan area (Portugal; 44 samples). To evaluate the real contribution of coffee powder to the mineral element content of the beverage, paired analysis of the water used in beverage preparation was also carried out.”
“The methodology section provides detailed descriptions of sample collection, preparation, and analysis.”
“The results are presented in an organized manner, with data on the concentrations of 20 elements in coffee and water samples. However, the interpretation is overly descriptive and lacks a critical analysis of trends or implications. Discuss the variability within samples and provide error margins or confidence intervals for key measurements. Highlight the implications of potentially toxic elements, such as lead and nickel, and their sources (e.g., water, coffee powder, or machine leaching).”
- The authors agree that the interpretation was overly descriptive. The discussion was updated with some considerations regarding the differences between the two studied countries and other available literature. Figure 3 was also updated to add error margins. Tables summarizing the data presented in the 3 figures were added as supplementary material in order to provide more detail to the interested reader.
“The discussion explores the mineral contributions of coffee powder and water, emphasizing the higher mineralization of Polish coffee. While informative, it lacks depth in exploring broader implications or limitations. Discuss how preparation variables (e.g., water temperature, pressure) might influence mineral extraction. Address study limitations, such as the small sample size or lack of control over preparation methods. Explore potential applications of the findings, such as optimizing brewing methods to minimize toxic element leaching. Compare findings with existing literature to contextualize the data.”
- The authors added some more detail to the discussion and compared the results obtained to other relevant literature. A few more sentences were added discussing the contribution of the consumption of the coffee espresso samples analysed in the present study to the daily intake of the macro- and trace elements. The authors also further discussed the implications of variations in the coffee espresso preparation procedure and the very composition of ground coffee could have on the concentration of the elements. However, because the authors did not collect any information regarding the coffee espresso preparation procedure (e.g., water pressure and temperature, amount of ground coffee used, etc.), origin or type of coffee, there really is not much that the authors can elaborate on this topic. Some interesting results regarding the relationship between the water and coffee concentration of specific elements was explored and elaborated in the discussion. A paragraph regarding the study limitations was also added.
“The figures and tables effectively summarize elemental concentrations and contributions from water. Use error bars or confidence intervals to show variability in the data.”
- Figure 3 has been updated with error bars.
Round 2
Reviewer 2 Report
Comments and Suggestions for Authors
It's ok.